

# Concentration Trajectory Route of Air pollution with an Integrated Lagrangian model (C-TRAIL model v1.0) derived from the Community Multiscale Air Quality Modeling (CMAQ model v5.2)

Arman Pouyaei [1], Yunsoo Choi [1], Jia Jung [1], Bavand Sadeghi [1], Chul Han Song[2]
[1] Department of Earth and Atmospheric Sciences, University of Houston, Houston, TX, USA
[2] School of Environmental Science and Engineering, Gwangju Institute of Science and Technology (GIST), Gwangju, South
Korea

*Correspondence to*: Yunsoo Choi (ychoi6@uh.edu)



**Abstract.** This paper introduces a reliable and comprehensive Lagrangian output (Concentration Trajectory Route of Air pollution with Integrated Lagrangian model, C-TRAIL version 1.0) from an Eulerian air quality model for validating the source-receptor link by following real polluted air masses. To investigate the concentrations and trajectories of air masses simultaneously, we implement the trajectory-grid (TG) Lagrangian advection scheme in the CMAQ (Community Multiscale Air Quality) Eulerian model version 5.2. The TG algorithm follows the concentrations of representative air "packets" of species along trajectories determined by the wind field. The generated output from C-TRAIL accurately identifies the origins of pollutants. For validation, we analyzed the results of C-TRAIL during the KORUS-AQ campaign over South Korea. Initially, we implemented C-TRAIL in a simulation of CO concentrations with an emphasis on the long- and short-range transport effect. The output from C-TRAIL reveals that local trajectories were responsible for CO concentrations over Seoul during the stagnant period (May 17-22, 2016) and during the extreme pollution period (May 25-28, 2016), highly polluted air masses from China were distinguished as sources of CO transported to the Seoul Metropolitan Area (SMA). We conclude that long-range transport played a crucial role in high CO concentrations over the receptor area during this period. Furthermore, for May 2016, we find that the potential sources of CO over that SMA were the result of either local transport or long-range transport from the Shandong Peninsula and, in some cases, from north of the SMA. By identifying the trajectories of CO concentrations, one can use the results from C-TRAIL to directly link strong potential sources of pollutants to a receptor in specific regions during various time frames.

**Keywords:** C-TRAIL, Trajectory analysis, CMAQ, East Asia, KORUS-AQ campaign





## 1    Introduction

Determining the long-range transport (LRT) of pollutants has been a challenge for air quality researchers over the years. As the chemical composition of outflow over a region or continent could significantly affect air quality downwind, information about LRT must be reliable. Several studies have applied a number of methods to examine the role that LRT plays in the concentrations of particulate matter (PM), ozone, different trace gases, and biomass burning tracers over target regions (Stohl, 2002). For instance, several sources (Choi et al., 2014; Lee et al., 2019; Oh et al., 2015; Pu et al., 2015) have applied the NOAA Hybrid single-particle Lagrangian integrated trajectory (HYSPLIT) model (Draxler, 1998) and back-trajectory analyses in an attempt to identify possible sources of PM in East Asia. The HYSPLIT model is a widely used tool that has been incorporated into other chemical-transport models (CTMs) to measure the LRT of ozone, carbon monoxide (CO), and aerosols to establish the source-receptor relationship of air masses over the United States (Bertschi and Jaffe, 2005; Carroll et al., 2008; Gratz et al., 2015; Price et al., 2004; Weiss-Penzias et al., 2004). Several studies have used another model, the FLEXTRA trajectory model (Stohl, 1996; Stohl and Seibert, 1998), to capture the background source regions of high-PM over East Asia and quantify the contributions from these regions (Lee et al., 2011, 2013). Furthermore, this model also has been applied to some European regions to explain the potential advected contribution of aerosols (Cristofanelli et al., 2007; Petetin et al., 2014; Salvador et al., 2008). Several studies have recently attempted to develop new trajectory models that overcome truncation errors that originate from numerical schemes in integrating trajectory equation (Döös et al., 2017; Rößler et al., 2018) and to link trajectories to specific trace species (Kruse et al., 2018; Stenke et al., 2009). Another widely used tool for studying the distribution of CO, ozone, PM, and other aerosols for both air quality forecasting and emission scenario analysis is the EPA's Community Multiscale Air Quality (CMAQ) model (Byun and Schere, 2006). CMAQ, with the help of meterological inputs by Weather Research and Forecasting (WRF) model, assists policy-makers with solving pollution-related issues by legislating regulations. Spatial concentration patterns of pollutants incorporated with other models (i.e., back-trajectory models) or satellite data enhances our understanding of the impact of LRT and other related processes such as the formation of aerosols, emissions, and dry deposition in various regions (Chen et al., 2014; Chuang et al., 2008, 2018; Wang et al., 2010; Xu et al., 2019; Zhang et al., 2019).

The conventional way of estimating potential source regions of air-mass transport is to use back-trajectory modeling. Frequently used for source-receptor linkage, it combines its output with measurements of pollutant concentrations. As this source-receptor linkage approach uses meteorology-based models for back-trajectories, it is not widely accepted because it is unable to determine whether an originated air mass is polluted or non-polluted (Lee et al., 2019). Thus, back-trajectory modeling provides unreliable information from which to assess the variation of pollutants at the receptor point, raising concern about using it for interpreting the contribution of the LRT effect to the concentrations of a target pollutant. In addition, other factors such as emissions and the local production of air pollutants contribute to variation in a target pollutant. Although aircraft campaigns in several regions have applied in a Lagrangian approach to interpreting variations in concentrations, they have not effectively addressed the above concern. After all, such campaigns are neither frequent nor continuous.





In this study, we implement a Lagrangian advection scheme that we refer to as the Trajectory Grid (TG) (Chock et al., 1996)
into the Eulerian CMAQ v5.2 model. We introduce a new type of output from the Concentration Trajectory Route of Air
pollution with the Integrated Lagrangian (C-TRAIL v1.0) stand-alone model in addition to CMAQ v5.2 output to
simultaneously accomplish two objectives: (1) to provide a direct link between real polluted air masses from sources and a
receptor and (2) to provide the spatial concentration distribution of several pollutants to explain relevant physical processes.
Chock et al. (2005) incorporated the TG into an air quality model to study the accuracy of this Lagrangian advection method
over the Bott advection scheme applied in the Eulerian domain. One significant outcome from the TG model applied to CTMs
is the ability to account for the concentration of pollutants in air masses to investigate the trajectory, which could address the
unreliability of meteorology-based Lagrangian models when the pollutedness or cleanliness of an originated air mass becomes
an issue. For this study, we have selected CO. As this pollutant has an oxidation lifetime of approximately two months, it an
ideal tracer with which we can study its impact on LRT without having stable background levels such as $CO_2$ (Heald et al.,
2003; Liu et al., 2010; Vay et al., 2011). Furthermore, as CO is produced mainly by the incomplete combustion of carbon-
containing fuels (Halliday et al., 2019), it is an ideal proxy with which we can relate concentrations of receptors to sources of
traffic or power-plant emissions. We begin by introducing the methodology behind TG and the implementation of TG into
CMAQ. Then, we present a simple case and our interpretation of the C-TRAIL output. Finally, we present a case study of C-
TRAIL for the Korea and United States Air Quality (KORUS-AQ) campaign over South Korea.
**2    Methodology**
**2.1    Trajectory grid approach description**
To solve the transport equation, Chock et al. (1996) presented the trajectory grid (TG) approach in air quality modeling. This
approach, which entails transporting points on a concentration profile along their trajectories in a Lagrangian manner, uses the
Eulerian approach for diffusive transport. TG method rewrites the advection equation for concentration as below:

$$\frac{dC}{dt} = \frac{\partial C}{\partial t} + \mathbf{v} \cdot \nabla C = -(\nabla \cdot \mathbf{v})C \tag{1}$$


where $C$ is the concentration of species in a velocity field $\mathbf{v}$. The Lagrangian approach divides the total derivative of
concentration into the full derivative of concentration with respect to time, $\frac{dC}{dt}$, and a remaining term containing velocity
divergence, $-(\nabla \cdot \mathbf{v})C$. Following this approach, the TG automatically and accurately conserves the mass, sign, and shape of
the concentration profile. As interpreted from the equation, the concentration profile of the species along trajectories can be
described. Otherwise stated, after determining the location of a packet and the concentration inside the domain, the
concentration profile along its trajectory could be assessed. Since all the species represented in one packet and all of the packets
move in the flow field according to the wind velocity, differentiating between advection equations for each species (as it is
done in Eulerian advection schemes) is no longer necessary. This will remove the associated numerical errors with a





discretization of the advection equation. The concentration of each packet along its trajectory can be determined by the following equation:

$$C(t) = C(t_0) \, exp\left(-\int_{t_0}^{t}(\nabla \cdot v)dt\right) \approx C(t_0) \, exp[-(\nabla \cdot v)(t - t_0)], \tag{2}$$

where $C(t)$ is the concentration of species at the location of a packet as it moves along its trajectory. Since the method can calculate the concentration from an ordinary differential equation, the method itself is mass conserving, monotonic, and accurate. However, in the diffusion step, interpolation errors occur but they are typically considerably smaller than Eulerian advection errors (Chock et al., 2005). In addition, the trajectory will be three-dimensional and as accurate as the input for wind velocity and direction. In particular, for large-scale vertical winds where typically CTMs modify the scheme to address the mass-conservation issue, TG will remove numerical diffusion from upwind vertical advection schemes and generate more physical vertical winds (Hu and Talat Odman, 2008). It is noteworthy to mention that units for the concentration of species are referred to as [ppbv] or [$\mu gm^{-3}$] depending on the species type, and the unit conversion is taken into account in the process of solving equations.

## 2.2    Implementation of TG in CMAQ v5.2

In this section, we briefly describe the key features of TG implementation in the CMAQ v5.2 model, an Eulerian model consisting of several modules (i.e., advection, diffusion, cloud and aqueous-phase). The C-TRAIL v1.0 model utilizes the same meteorology, initial conditions (ICs), boundary conditions (BCs), and emissions that CMAQ requires. All of the CMAQ modules and parameters are associated with cells of the Eulerian grid on the model domain. Since TG is based on CMAQ in this study and some of the CMAQ processes cannot be satisfactorily carried out by Lagrangian models (e.g., eddy diffusion) at this time, grid cells are the primary structure for initiating and listing packets. By grouping, the packets in grid cells, keeping track of which packets are close to each other, is also easier. While the grid cells of Eulerian models represent Eulerian-type outputs, tracking the packets of Lagrangian advection will provide their trajectories in addition to their concentrations (Figure 1).

The process of advection for packets follows the ordinary differential equation:

$$\frac{d\mathbf{y}(t)}{dt} = \mathbf{V}(\mathbf{y}(t), t) \tag{3}$$

where $\mathbf{V}$ [ms$^{-1}$] is the three-dimensional wind velocity and $\mathbf{y}(t)$ [m] is the position vector of packets at time $t$ [s]. The equation is solved using the following simple predictor-corrector scheme:

$$\mathbf{y}^i(t + \Delta t) = \mathbf{y}(t) + \mathbf{V}(\mathbf{y}(t), t)\Delta t \tag{4}$$

$$\mathbf{y}^f(t + \Delta t) = \mathbf{y}(t) + 0.5[\mathbf{V}(\mathbf{y}(t), t) + \mathbf{V}(\mathbf{y}^i(t + \Delta t), t + \Delta t)]\Delta \boldsymbol{t}, \tag{5}$$





where $\mathbf{y}^i$ is the initial estimate of the new position from the predictor step and  $\mathbf{y}^f$ is the final position calculated by the
corrector step. When the initiated packets in the domain follow the Lagrangian equation, they land in different grid cells after
each time step. To balance the density of packets in grid cells, we apply a simple packet management technique that includes
spawning (filling) and pruning (emptying) processes. In the spawning process, every step entails the creation of a group of
new packets in each cell that has too few packets.  The initial composition of a spawned packet is estimated from nearby
packets. The pruning process entails the removal of extra packets from cells that have become overpopulated. During this
process, the packets closest to the cell center are retained. Such packet management with favorable options contributes to
reducing the computational costs of the C-TRAIL model. The underlying algorithms for both vertical and horizontal diffusion,
emissions, and other processes are the same as those in standard CMAQ (Byun and Schere, 2006) with some minor
modifications. The coupling of Eulerian diffusion and TG advection at each time step is accomplished by first taking the
concentration average from all the packets in each cell as cell average. Then, by considering each packet as its cell and cell
averages representing the neighboring cells, we used a predictor-corrector method for determining each packet's concentration.
Figure 2 summarizes the process of C-TRAIL from initialization to output generation. By combining the location of each of
the unique packets in each time-step during the 24 hours, we generate the 24-hr trajectory of each packet.

## 3    Models' setup and validation

In this study, we implement TG in the CMAQ model version 5.2. The model domain with a horizontal grid resolution of 27-
km over East Asia covers the eastern parts of China, the Korean Peninsula, and Japan, shown in Figure 3. We use the 2010
MIX emission inventory (Li et al., 2017) at a 0.25-degree spatial resolution. This emission inventory contains monthly
averaged carbon bond version 5 (Sarwar et al., 2012) emission information that includes ten chemical species, including CO
in five different sectors.  We also use the 2011 Clean Air Policy Support System emission high-resolution (1-km) inventory
from the National Institute of Environmental Research for Korea, which contains the area, and the line and point sources of a
variety of species, including CO.  We provide WRF model v3.8 output as meteorological inputs in our CMAQ model. Jung et
al. (2019) validated the model setup, in a comparison between aerosol optical depths from simulations and observations, they
showed a correlation of 0.64 for the entire KORUS-AQ campaign period. Their comparison of various gaseous and particulate
species also showed close agreement with observations.
We run C-TRAIL simulations for May 2016 during the KORUS-AQ campaign. In papers pertaining to this campaign, several
studies separated the time frame into three periods (Table 1) based on meteorological conditions: 1) the dynamic weather
period (DWP), a rapid cycle of clear and rainy days in the Korean Peninsula (May 10-16); 2) the stagnant period (SP), in which
the area was under the influence of a high-pressure system (May 17-22) and which showed the influence of local emissions;
3) the extreme pollution period (EPP) with high peaks of pollutants that showed strong direct transport from China (May 25-

143   28).



The overall accuracy of CMAQ CO simulation compared to that of aircraft measurements during all periods is presented in
Figure 4(a). The correlation between the modeled CO concentration and the observations through different altitudes for the
entire May 2016 was 0.71, indicating that the performance of the model is sufficiently reliable for a study of the source of CO
concentration (Table 1). Figure 4 shows the under-prediction of the model during the DWP and SP. The model, however,
showed a very high correlation during the EPP compared to higher CO observations over the Korean Peninsula. The C-TRAIL
outputs of the mentioned periods will be discussed in Section 3.3.

The Eulerian output from CMAQ, including CO concentrations and surface wind fields, is displayed in Figure 5. Southeastern
China, including the Shanghai region and the Shandong Peninsula, showed a high peak of CO concentrations because of their
high anthropogenic emissions. The impact on pollution of LRT is greater in this region because the dominant wind in May is
westerly over East Asia, which explains why high concentrations of CO are observed over the Yellow Sea. Also observed over
the Yellow Sea during this period is a shallow anticyclone, a common phenomenon in this region that affects the regional
transport of pollution. By conducting a thorough investigation of CO concentration and wind patterns during various
meteorological periods, we achieved the following highlights: 1) During the DWP, a mixed response from the LRT of CO and
local emissions occurs. Also, in light of the impact of convection, the concentrations of CO over Korea, whether high or low,
depends on the region of the CO sources. Owing to the dynamic nature of this period (i.e., cloudy, rainy, or clear), the
interpretation of the LRT effect by conventional methods is challenging. 2) During SP, a high-pressure system settles over the
Korean Peninsula, which explains the extremely low wind speed, and the stagnant air, the latter of which will eliminate the
impact of LRT. Even though reducing the complexity of the atmosphere may make the model perform better simulations, the
CO concentration is highly underestimated by the model (Jeon et al., 2016) due to uncertainties and/or faulty emission
inventories over East Asia. 3) During EPP, as it is shown in Figure 5, the anticyclone over the Yellow Sea contributes to the
transport of more CO from China to the Korean Peninsula. Furthermore, high concentrations of CO in regions throughout
China are observed. Thus, a combination of these two effects would lead to model predictions of higher concentrations over
Korea.
The raw hypothesis from Eulerian outputs is that a high CO concentration at a receptor during a specific period is due to LRT
from a source because the average wind moves toward the receptor during that period. This hypothesis is based on the average
wind speed and direction and the average CO concentration, which do not constitute a reliable source of this assumption. We
will briefly explain why we require merged output with simultaneous changes in trajectories and concentrations. To determine
the source of LRT, researchers should include one major parameter in their investigations: the trajectory of the air mass. Once
the location of the source and the trajectory of the air mass is known, the air mass is assumed to be polluted. If the air mass is
not polluted, then that source is not responsible for high concentrations in the receptor location. Therefore, linking the source
to the receptor based on only mean wind patterns and concentrations is not a reliable approach. The following paragraphs will
discuss how we combine concentrations and trajectories into one set of outputs to better explain the trajectories.





## 4    C-TRAIL analysis

Because C-TRAIL is a derivative of CMAQ, both a Lagrangian output and CMAQ's standard Eulerian output will be available after each run. The C-TRAIL will help us identify source-receptor linkage, save the full trajectory of packets, and display the path of selected packets. Therefore, not only will C-TRAIL simulations provide all spatial concentration changes, but they will also display the trajectories of each packet, owing to the Lagrangian approach of TG.  In addition, we will be able to determine changes in concentrations along this trajectory. The difference between this model and other meteorological-based models is that they enable us to study changes in the concentrations of selected species along different paths, investigate the evidence for the amount of pollution in originated air masses, study the reason behind the oscillation of concentration, and the linkage of oscillations to both sources and sinks along the path.

In this section, we will provide an example of how we use C-TRAIL to study the sources of different packets from different altitudes over the Seoul Metropolitan Area (SMA), and in later sections, we will focus on the entire month of May 2016 C-TRAIL over the SMA. Figure 6 shows the C-TRAIL output for June 4, 2016.  We gathered all of the packets over the city of Seoul and analyzed the trajectory of each packet . Figure 6(a) shows the path of all the packets from their sources.  Packets are represented by different colors. It is observed that some of the packets came from southeastern South Korea, and one of them originated in southeastern China, moved over the Yellow Sea, and landed in Seoul. Some of the packets also originated northwest of South Korea from northern China. Most of the packets, however, were locally initiated, generally from regions around the SMA. Relatively similar trajectories found when using HYSPLIT back-trajectory model (Fig. S1).

Figure 6(b) depicts how the CO concentration of the four most aged packets changes as they travel on their path toward Seoul. This type of output is a new feature and has not been studied before. With meteorological-based back-trajectory models, the path of air parcels and their back trajectories could be delineated; we are the first, however, to study the concentrations of species (in this case, CO) through the path of air packet using a CMAQ-based Lagrangian integrated model. The four most aged packets came from 24, 21, 20, and 17 times step back (Figure 6(b)). We find these packets interesting because they follow a long path, changes in their concentrations fluctuate, and they are easy to comprehend. From studying these packets and their C-TRAIL, we generally understand that the concentration of each packet increases as it approaches the SMA. The concentrations of near-surface packets tend to fluctuate more than those of the high-altitude packets. Also, larger oscillations in the concentrations occur over land rather than the ocean. This impact, however, becomes more vivid when a near-surface packet reaches land from the ocean and reaches a sudden peak in concentration.  The sudden peaks of the concentrations of the near-surface packets are due to their movement over either a city or some source of emissions. Over the SMA and other cities, two peaks, mainly caused by on-road emissions, occur during local morning and evening times.

## 5    Case Study for C-TRAIL analysis: May 2016 KORUS-AQ period

Using a conventional method with model data gathered over the course of a month or a year to incorporate concentrations into a trajectory analysis produces a tremendous amount of outputs, making it difficult to interpret all the outputs simultaneously.





For our case study, covering May 2016, we selected Seoul, South Korea over East Asia as the receptor. We plotted C-TRAIL
outputs according to variations in the packet concentrations and their distances from the receptor. Figure 7(a) presents the
general path of all packet trajectories reaching Seoul at 9:00 AM LT throughout May 2016. The color bar represents the altitude
at which the packets have been traveling. Generally, packets at low altitudes travel from local areas to Seoul, and those at high
altitudes travel from more distant regions. One exception was packets that originated in the Shandong Peninsula; some traveled
at high altitudes and some at low altitudes. Figure 7(b) displays a C-TRAIL that represents a unique type of packet that follows
the concentrations of trajectories. In this case, each packet at each location (or hour of the trajectory) has a specific CO
concentration that depends on its altitude (high altitude/surface), location (land/sea/urban/forest), and hour of the day (traffic
hours/non-traffic hours). To better explain the location of packets and variability in their trajectory paths before reaching Seoul,
we created a boxplot, shown in Figure 7(c), of packet distances in kilometers from the receptor at each hour before the packets
reach Seoul. When the packets reach Seoul at 9:00 AM local time, the distance becomes zero. Furthermore, the boxplot of
trajectories' heights for all the periods is presented in Fig. S2. In a study of the C-TRAIL outputs, it is better to take into
account the trajectories, concentrations, and distances simultaneously. Considering this note, the concentrations and distances
of packets at early hours (10:00 AM to 2:00 PM of local time) in Figure 7 show high variability in concentrations with a
median of around 150 ppbV and a maximum as high as 500 ppbV. Most of these packets originated far from the receptor (i.e.,
eastern, northern, and southeastern China). The median of the concentration shown in the boxplot rose slightly between 6:00
PM and 10:00 PM. Also, distances showed more variations during this period, which can be explained by the different paths
of the trajectories (i.e., local trajectories with shorter distances and LRT trajectories with longer distances). As the packets
approach Seoul (6:00 AM to 9:00 AM), the upper whisker of concentration values increased to as high as 400 ppbV, and the
distances approached zero, indicating higher concentrations of CO over local trajectories due to surface on-road emissions and
other emission sources.
For the DWP because of variable weather and wind (i.e., cloudy, rainy, or clear), C-TRAIL showed a mixed response of the
trajectories from both local and long-range transport, shown in Figure 8(a). A wide interquartile range and a median of close
to the 25th percentile at 11:00 AM and 12:00 PM indicate that a few packets contained high concentrations of CO (close to
300 ppbV), but the majority consisted of low concentrations (around 100 ppbV). The distance output of low-concentration
packets showed distances as long as 500 km (over the Shandong Peninsula). As the packets approached Seoul, the median
concentration values were as high as 150 ppbV. Thus, from Figure 8, we conclude that most of the long trajectories followed
a path at high altitudes (higher than 7 km), and the polluted trajectories, which originate in the Shandong Peninsula, are from
near-surface shown in Figure 8(a).
Unlike the DWP, the SP showed a more vivid display of trajectories, nearly all of which could be considered local trajectories.
Long-range trajectories could not be considered responsible for the CO concentration value of Seoul. After all, from 10:00
AM to 4:00 PM (Figure 9(a) and (b)), nearly all of the long-distance packets show concentrations of less than 100 ppbV. The
local origination of highly polluted trajectories can be explained by a high-pressure system over the Korean Peninsula during
this period, which was responsible for very low wind speeds. The poor emission inventory over East Asia, however, provided





extreme under-predictions of high concentration values during this period. Therefore, when studying model outputs, we should
account for various aspects of the model (e.g., the transport, diffusion, formation, deposition, and convention), in which
diffusion plays a significant role in CO concentration values at the receptor location in this case.
During the EPP, several high concentrations of CO appear at the early points of trajectories. These high concentrations,
combined with high distance values, indicate that the LRT of polluted air masses is responsible for high concentrations of CO
during this period (Figure 9). Furthermore, the variability of CO concentrations from 10:00 PM to 9:00 AM at the receptor
location stems from both the various paths of the trajectories and distances. Along the high concentration trajectories close to
the surface, which originate in the Shandong Peninsula, pass over the Yellow Sea and land in Seoul at 9:00 AM. When the
surface packets reach urban areas, they present maximum CO concentrations, depending on the time of day and rush-hour
traffic. The assumptions made by studies that used Eulerian model outputs or meteorological-based Lagrangian models for
this period were that transport played an important role(Lee et al., 2019).  The outputs from C-TRAIL also indicate that highly
polluted air masses originated in China (the source) and landed in Seoul (the receptor). That is, the findings regarding the
trajectories and the origin of polluted air masses are indisputable.
We further analyzed the diverse aspects of C-TRAIL results using the Open-air package in R (Carslaw and Ropkins, 2012)
and determined the frequency of trajectories passing through every one degree by one degree gridded area, illustrated in Figure
11(a). Central China, northern China, and North Korea are not common areas for packet movement because packets most
likely pass only once through the grids of these regions (at a frequency of about one percent). For the Yellow Sea and the
Shandong Peninsula region, however, trajectories are more likely to pass at around a frequency of ten percent.  The figure also
shows that most of the trajectories (25 to 100 percent) pass over the west side of the SMA, a two-degree by two-degree area
(the dark red section in Figure 11 (a)). We can classify trajectories into separate segments according to their concentrations.
Figure 11(b) shows this type of classification and the link between the average concentration of all trajectories to their paths.
While higher concentrations are most likely the result of local transport, lower concentrations are most likely from LRT. For
the May 2016 case, whereas most of the high concentration values correspond to packets that originated in South Korea or
close to SMA, most of the low concentration values correspond to packets originating in China but their impact is still there.
By clustering the outputs of C-TRAIL, we are better able to locate the dominant paths for the May 2016 trajectories. According
to Figure 12(a), based on the Euclidean distance function, about 37.8% of trajectories come from local areas to the east, south,
and north of the SMA. About 16.1% of trajectories originate in northern China and follow paths over the Yellow Sea to the
SMA; About 10.5% of trajectories come from southwestern South Korea over the Yellow Sea to reach the SMA; about 21.3%
of trajectories come from the Shandong Peninsula; and the remaining trajectories (5.3%) originate in central China and are
transported over China and the Yellow Sea to the SMA. Angle clustering in Figure 12(b), however, tells a different story about
the trajectories. Clustering by the angle distance function shows that the angles from the starting points of the back trajectories
may be similar. Generally, nearly all of the packets come from the west side of the SMA, 32.2% originate farther west, 34.5%
originate in the south-west, 12.7% come from the south/south-west, 14.9% come from the northwest and 5.7% come from
east/southeast. This clustering is consistent with strong westerly winds during the spring in East Asia.




By quantifying clusters based on their trajectories, cluster analyses show the relative importance of regional sources.
Nevertheless, they are not completely accurate at determining the relative contribution of potential source regions because they
do not consider the concentrations along with trajectories. One method for calculating the probability of potential sources is
the potential source contribution function (PSCF), which calculates the probability that a source is located at a specific latitude
and longitude (Pekney et al., 2006). Figure 12(c) shows that the probability of packets with high concentrations (i.e., those
with concentrations at or above 90 percentile) passing over the Yellow Sea and reaching the SMA from the southwest is higher
than 0.3.  Two areas through one packet containing a high concentration of pollutants passed show a high probability of 0.6
and 0.5. One is southwest of the SMA over the Yellow Sea and the other lies between North Korea and the coast of northern
China over the Yellow Sea.

One important limitation of the PSCF is that distinguishing between moderate and strong sources is difficult. To overcome
this problem, we can apply the concentration-weighted trajectory (CWT) method to compute concentration fields for
identifying strong source areas of pollutants. The CWT method, based on concentration values over each trajectory, estimates
the trajectory weighted concentration in each grid cell by averaging the sample pollutant concentrations of trajectories crossing
each grid cell (one degree by one degree). CWT shows close agreement with PSCF results. Figure 12(d) shows the distribution
of weighted trajectory concentrations of CO in May 2016 surrounding the SMA. The CWT results depict that not only could
the Yellow Sea and the Shandong Peninsula be potential sources of high concentration over the SMA, but other local sources
may also be considered strong sources.  For example, the Pyongyang area in North Korea has a high concentration, weighted
over 250 ppb, which indicates a strong potential source of CO in this month. Furthermore, local regions such as those to the
west, east, and south of the SMA show a strong potential source of high CO concentration in Seoul. Among the long-distance
sources, only the Shandong Peninsula and some parts of northern China have CO concentrations of around 100 ppm according
to the CWT analysis; other long-distance sources are not strong sources because of the scarcity of trajectories in these areas,
so we consider them rare sources. For instance, although the LRT showed to be the reason for high CO concentrations over
the SMA during extreme pollution period (May 25-28), but in longer periods (e.g., one month or one year), with that amount
of contribution, distant regions from the SMA may not be considered as potential strong sources.

## 6    Conclusion

In this study, we introduced Lagrangian output, C-TRAIL, extracted from the Eulerian CMAQ model. The C-TRAIL
comprehensive output directly linked trajectories of pollution from the source to the receptor. We used concentration and
trajectory values of C-TRAIL outputs to investigate the pollution status of originated air masses by classifying the outputs for
May 2016 over East Asia into separate categories.  Unlike the conventional Eulerian CO concentration plots for separate





periods, which did not exhibit a reliable and clear relationship between the source and the receptor, the C-TRAIL outputs,
which combined trajectories and concentrations, demonstrated the impact of LRT on pollution during the EPP accurately.
Furthermore, during the DWP, C-TRAIL outputs showed that polluted packets from the Shandong Peninsula were responsible
for high CO concentrations. The outputs for the SP revealed CO concentrations of less than 100 ppbV for distant packets,
strong evidence supporting the link between local trajectories and CO concentrations over the SMA during this period.
More comprehensive investigations on C-TRAIL outputs revealed that the Shandong Peninsula, local regions near the SMA,
and the Pyongyang area were potential strong sources of CO pollutants over the entire month of May 2016. Overall, by
analyzing the trajectory path of packets landed in a specific location, we were able to generalize that C-TRAIL represents an
ideal tool for addressing the impact of long-range transport on species concentrations over at receptor by providing
concentration and trajectories simultaneously. C-TRAIL can be applied to LRT-impacted regions such as East Asia, North
America, and India. Owing to the uncertainties inherent to emission inventories and immature diffusion methods in modeling,
however, C-TRAIL outputs may have limitations that we will address in future work. The objective of this study is to suggest
an effective tool for establishing a link between real sources of pollution to a receptor via trajectory analysis. The results of
this study over East Asia showed the reliability and various advantages of C-TRAIL output. Therefore, because of its capability
to determine trajectories of masses of CO concentrations with high computational efficiency, C-TRAIL output could prove to
be a highly efficient, useful tool for those modeling air quality over a specific region and investigating sources of polluted air
masses.

**Code Availability.** The C-TRAIL version 1.0 is available for non-commercial research purposes at
https://github.com/armanpouyaei/C-TRAIL-v1.0.

**Supplement.** The supplementary document related to this article is available.

**Author Contribution.** A.P., Y.C., and B.S. contributed to the design and implementation of the research. J.J. prepared the
CMAQ model and inputs. A.P. prepared the model, analyzed the results and took the lead in writing the manuscripts. Y.C. and
C.H.S. supervised the project. All authors discussed the results and commented on the manuscript and contributed to the final
version of the manuscript.

**Competing interest.** The authors declare no competing financial and/or non-financial interests in relation to the work
described.

**Acknowledgments.** We want to acknowledge Dr. Peter Percel for his technical help in the development of CMAQ-TG in this
research. This study was funded by the National Strategic Project-Fine particle of the National Research Foundation of Korea



(NRF) funded by the Ministry of Science and ICT (MSIT), the Ministry of Environment (ME), and the Ministry of Health and
Welfare (MOHW) (NRF-2017M3D8A1092022).



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




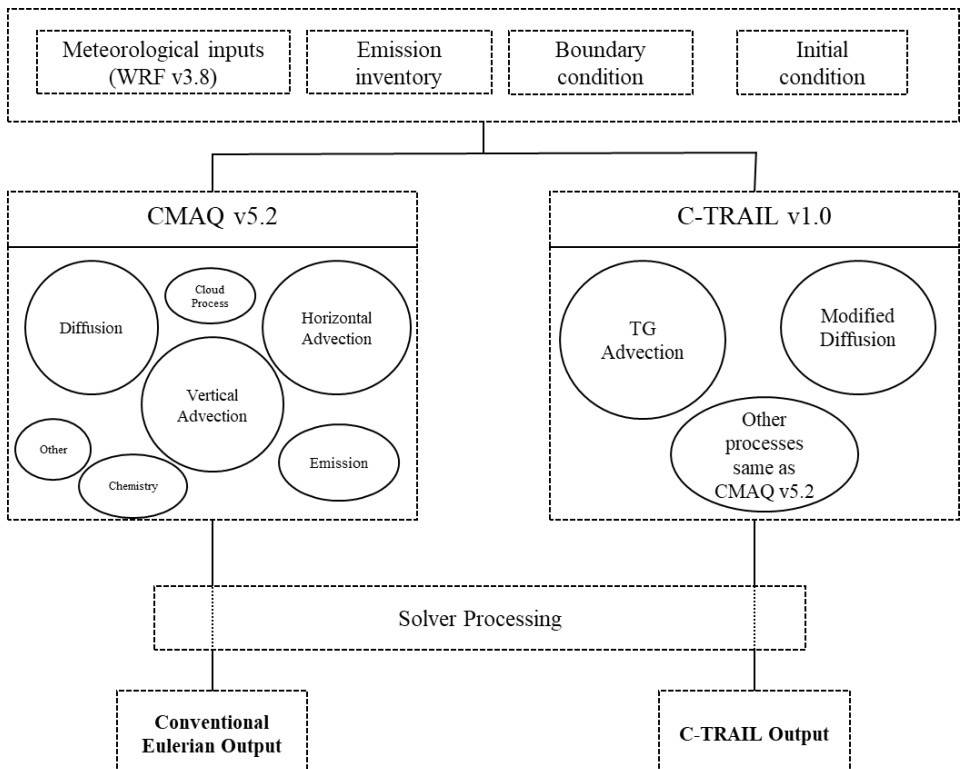


**Figure 1: Schematic of the conventional CMAQ output versus C-TRAIL**






1. **Initialization**

   - Packets are generated based on the IC and BC in grid cells.

   - Each packet receives an ID, x, y, z.

   - Time-step selection is based on the condition that a packet may not travel more than ¾ of the distance between all opposite faces of a grid cell.

2. **Packet Management**

   - The number of packets in each grid cell must not exceed five.

     *If* exceeded: Remove the extra packets.

     The concentration of remaining packets will be the average of all available packets.

   - Each grid cell must not be empty.

     *If* empty: Add an extra packet.

     The concentration of the added packet will be similar to that of the closest packet.

   - The properties of the packets are controlled through all time-steps.

3. **Advection**

   - The three-dimensional advection equation is solved to update the location of packets.

4. **Diffusion**

   - The implicit Eulerian diffusion equation is solved to obtain an average over the number of packets in each grid cell.

   - By considering each packet the center of the cell and the cell average as neighboring cells, the diffusion equation is solved for each packet using the predictor-corrector method.

   - Horizontal diffusion requires extra sub-grid diffusion for pair-wise diffusion.

5. **Emission**

   - The emission fluxes of various species (similar to CMAQ) are added to each packet through vertical diffusion.

6. **Output Generation**

   - Based on Lambert's projection, the x, y, and z of each packet are converted into longitude, latitude, and altitude.

   - C-TRAIL is generated during every output time-step (1 hour).

**Figure 2: Algorithm of the CMAQ-TG model**






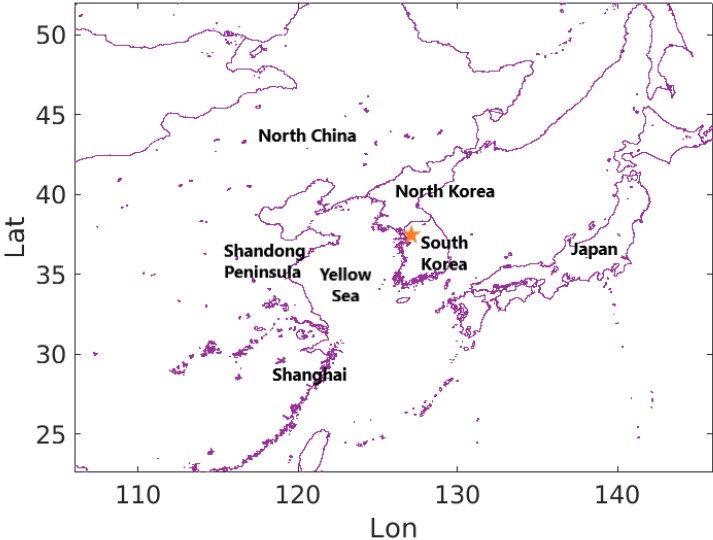


**Figure 3: Domain of the study; the orange star indicates the Seoul Metropolitan Area (SMA)**







**Table 1: Comparison of the statistical parameters of CMAQ CO concentrations and aircraft measurements (COR: correlation, IOA: index of agreement, RMSE: root mean square error, MAE: mean absolute error)**

|  |  | Abbreviation | COR | IOA | RMSE | MAE |
|---|---|---|---|---|---|---|
| a) | Entire month of May 2016 |  | 0.71 | 0.72 | 91.3 | 66.7 |
| b) | Dynamic Weather Period | DWP | 0.72 | 0.62 | 81.5 | 66.2 |
| c) | Stagnant Period | SP | 0.65 | 0.58 | 98.4 | 83.3 |
| d) | Extreme Pollution Period | EPP | 0.89 | 0.88 | 68.7 | 47.7 |








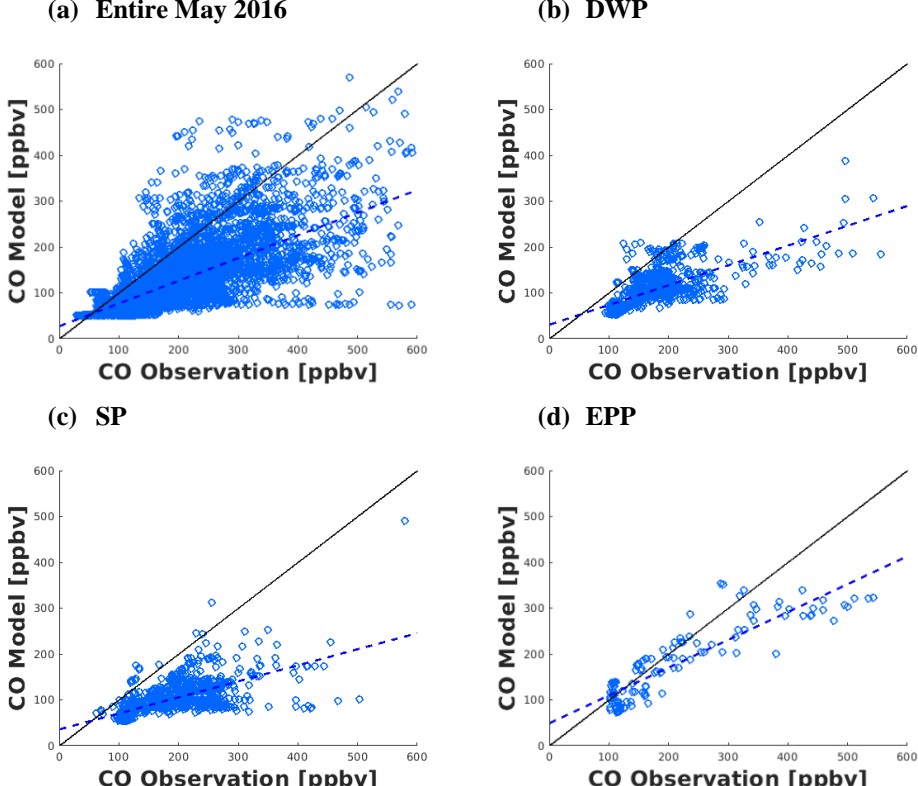

**Figure 4: CMAQ model results versus aircraft CO measurements for (a) all of May 2016,**
**(b) the DWP, (c) the SP, and (d) the EPP**



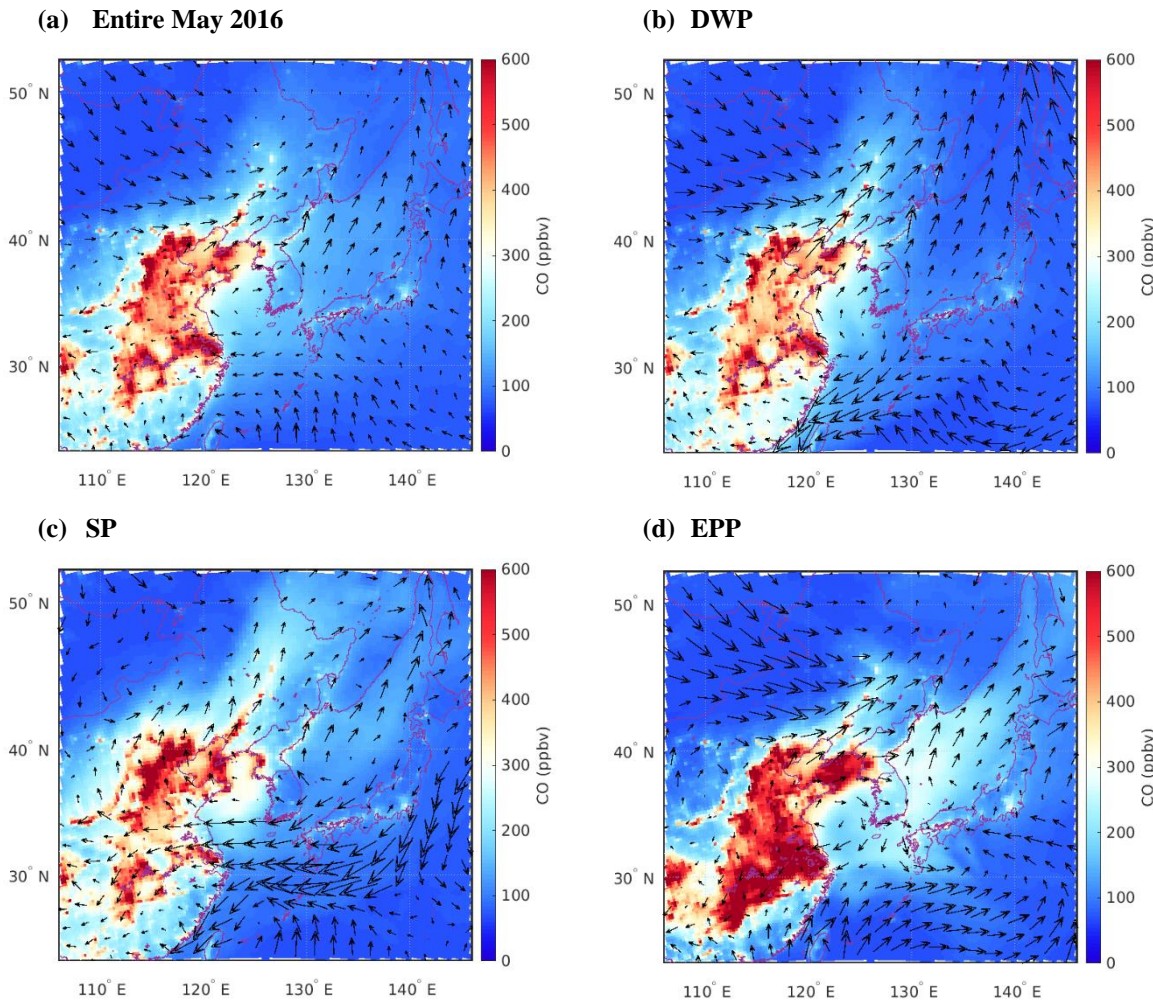

**Figure 5: Model CO concentrations and wind patterns over the surface during (a) all of May 2016,**
**(b) the DWP, (c) the SP, and (d) the EPP**


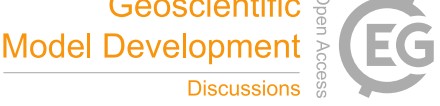


(a)

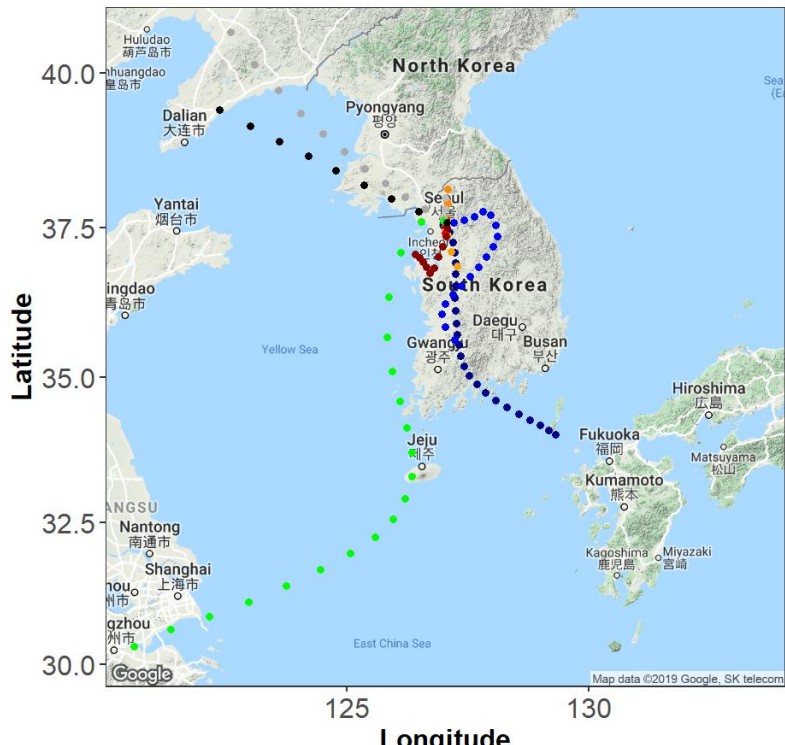

(b)

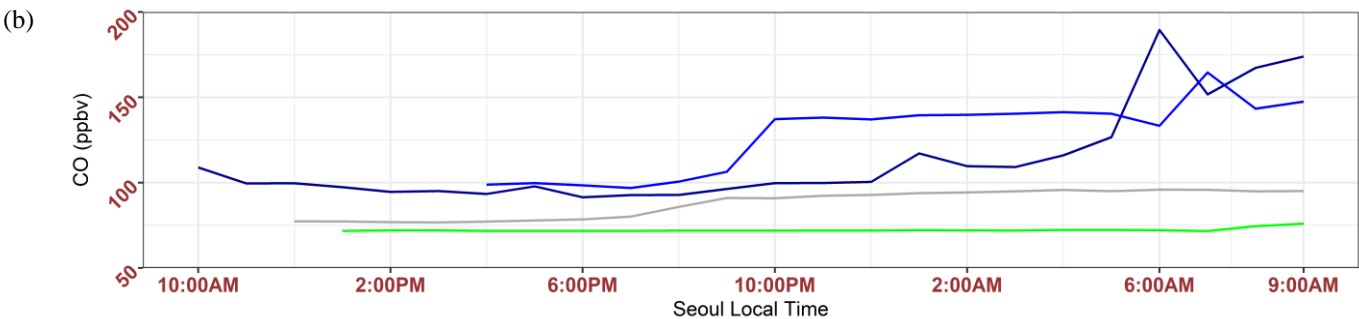

**Figure 6: C-TRAIL output for June 4, 2016: (a) trajectory of packets reaching Seoul at 9:00 AM local time (b) changes in the CO concentration of four aged packets moving toward Seoul from source points**






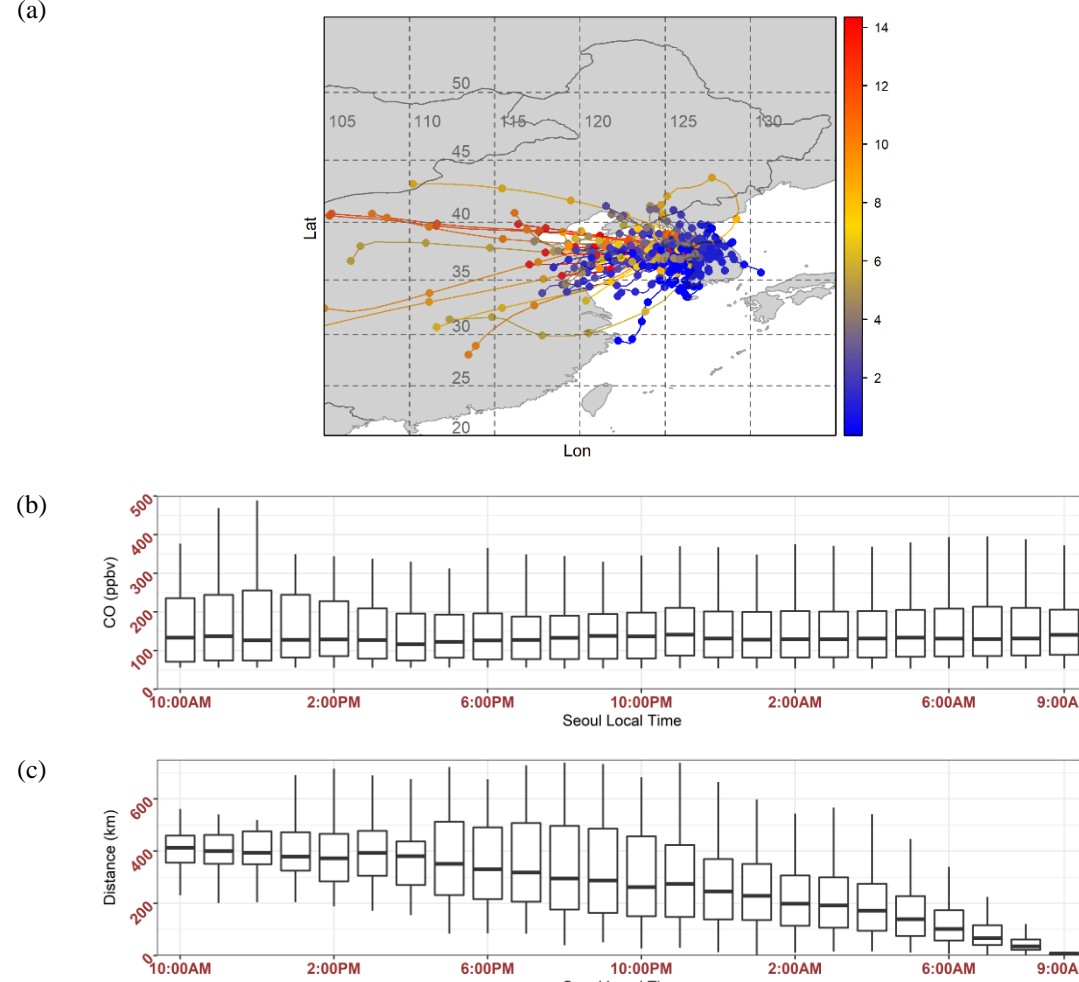

**Figure 7: C-TRAIL output for the entire month of May 2016 for Seoul, the receptor: (a) 24-hour trajectory of packets at various**
**altitudes, (b) the boxplot of the CO concentrations of all packets at each hour before they reach Seoul, and (c) the boxplot of**
**packet distances from Seoul every hour before the packets reach Seoul**






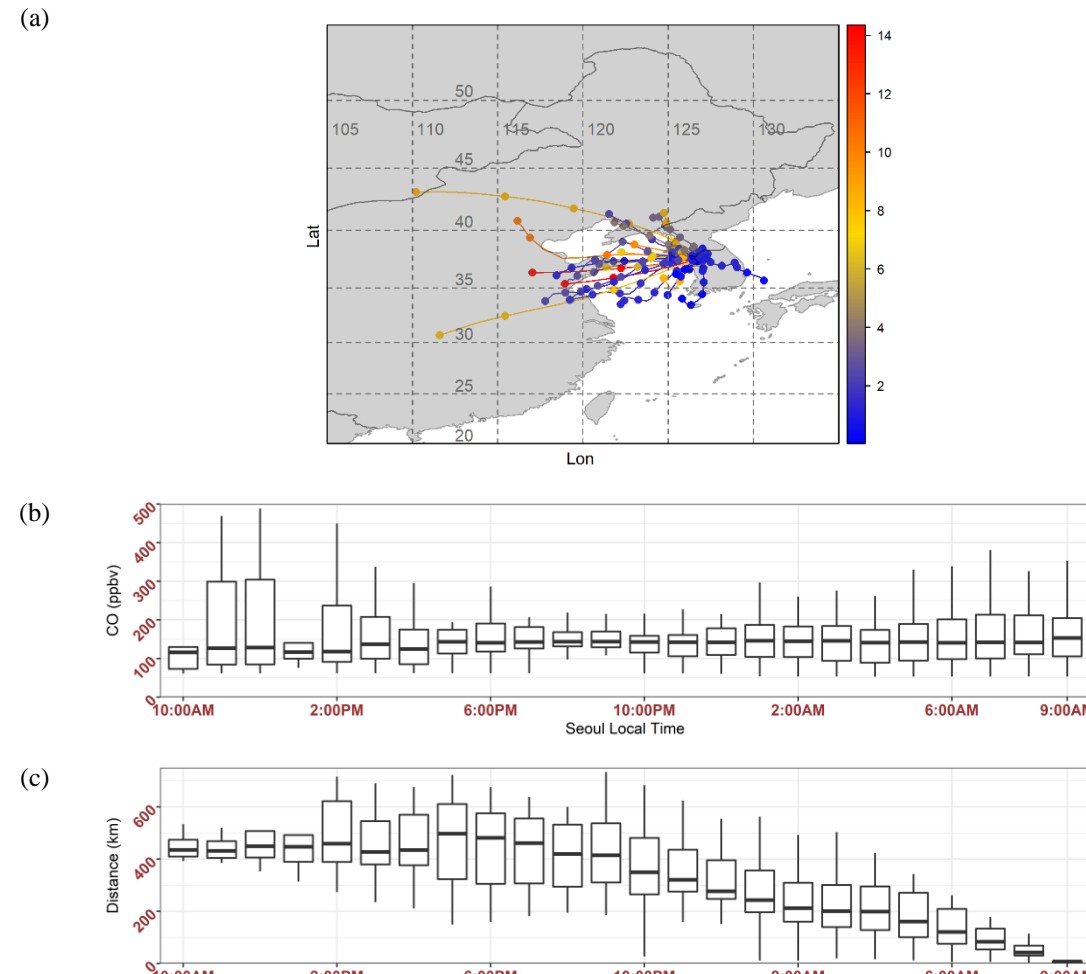

**Figure 8: C-TRAIL output for the dynamic weather period (DWP) for Seoul, the receptor: (a) 24-hour trajectory of packets at**
**various altitudes, (b) the boxplot of the CO concentrations of all packets at each hour before they reach Seoul, and (c) the boxplot**
**of packet distances from Seoul every hour before the packets reach Seoul**




(a)

(b)

(c)

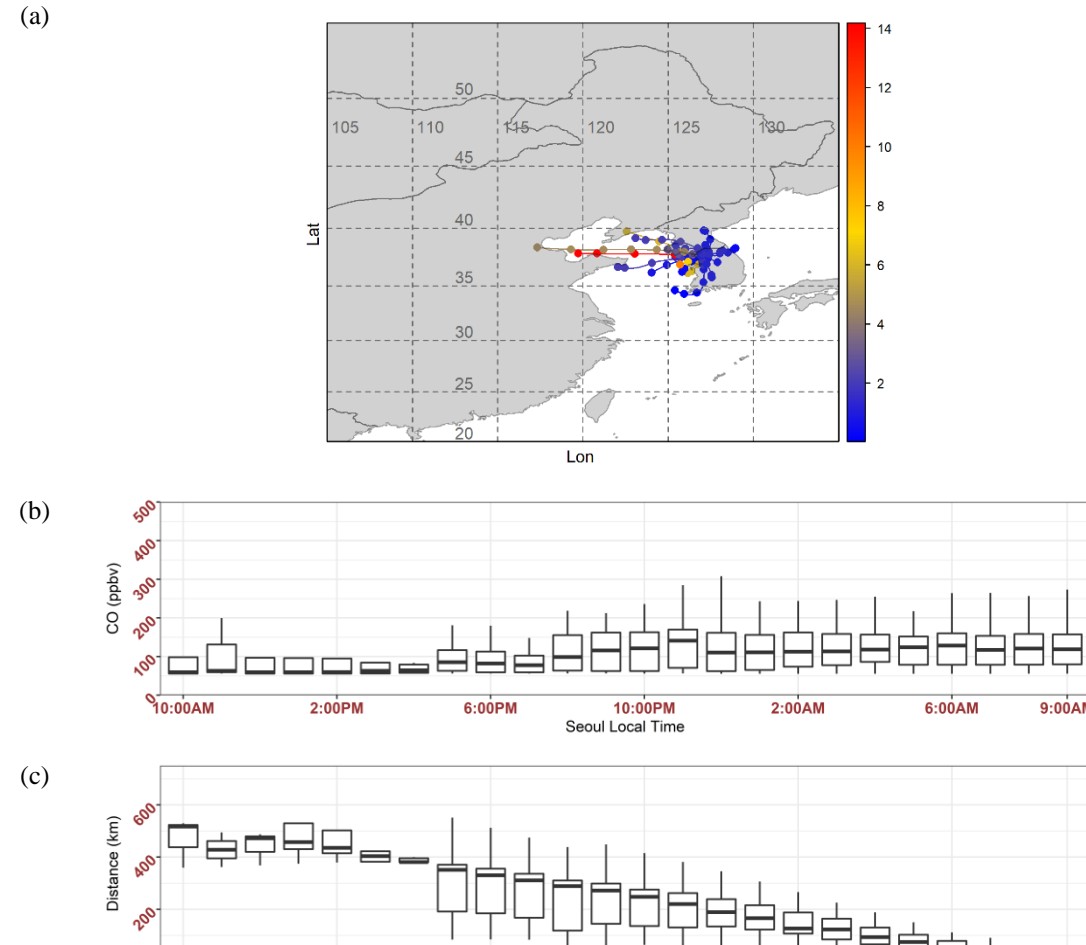

**Figure 9: C-TRAIL output for the stagnant period (SP) for Seoul, the receptor: (a) 24-hour trajectory of packets at various altitudes, (b) the boxplot of the CO concentrations of all packets at each hour before they reach Seoul, and (c) the boxplot of packet distances from Seoul every hour before the packets reach Seoul**




(a)

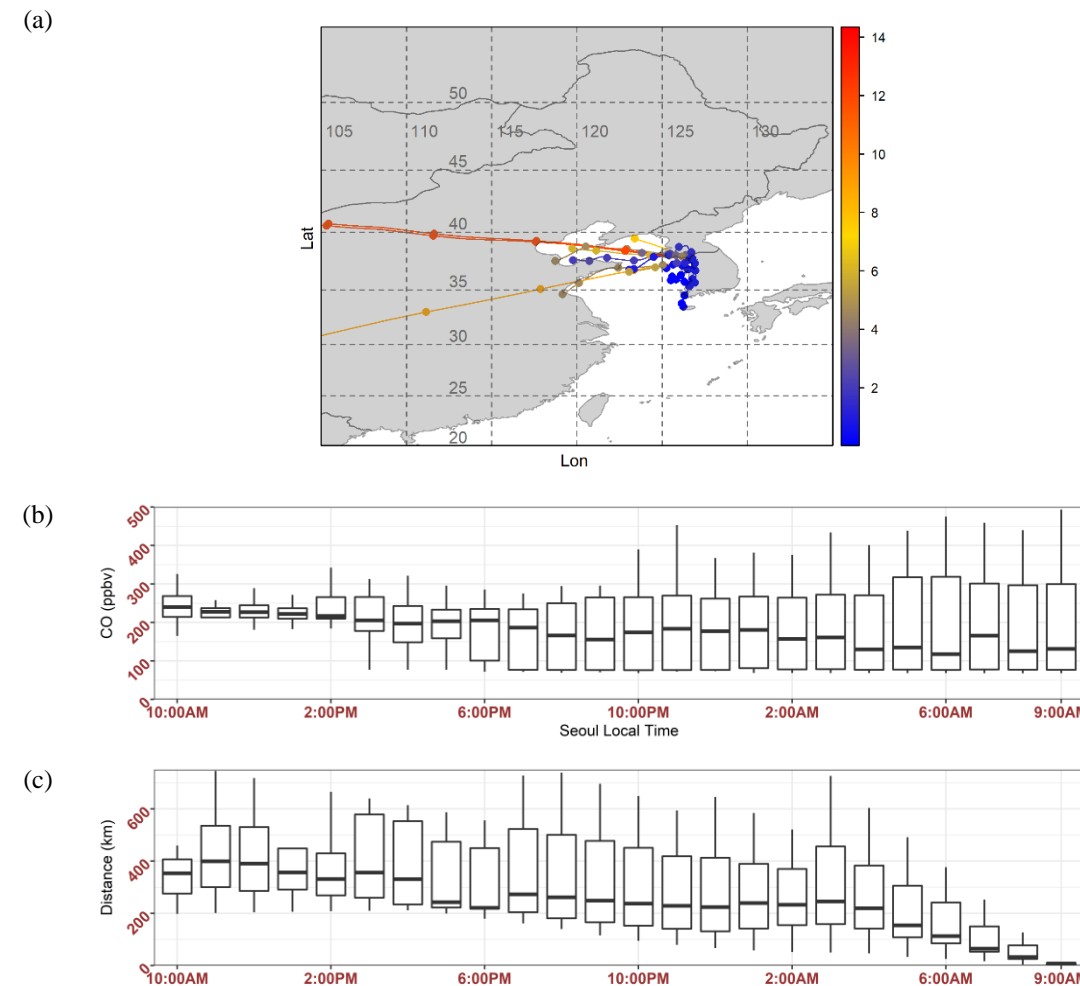

(b)

(c)

**Figure 10: C-TRAIL output for the extreme pollution period (EPP) for Seoul, the receptor: (a) 24-hour trajectory of packets at various altitudes, (b) the boxplot of the CO concentrations of all packets at each hour before they reach Seoul, and (c) the boxplot of packet distances from Seoul every hour before the packets reach Seoul**







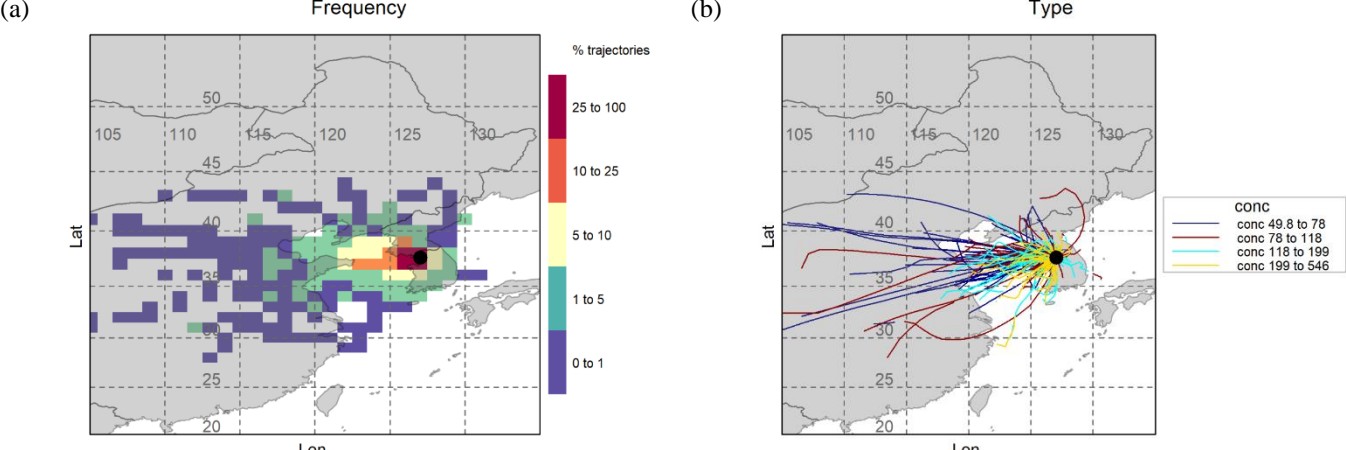

**Figure 11: (a) Plot of the frequency of trajectories and (b) the trajectories, classified by their concentration values**


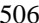


**Figure 12: (a) Trajectories clustered by the Euclidian distance function, (b) trajectories clustered by the angle distance function, (c) the potential source contribution factor plot, and (d) the concentration-weighted trajectory plot**

