# Peer review of "Concentration Trajectory Route of Air pollution with an Integrated Lagrangian Model (C-TRAIL Model v1.0) Derived from Community Multiscale Air Quality Model (CMAQ Model v5.2)"

_Geoscientific Model Development, 2019_

## Referee Comment (RC1) · Anonymous Referee #1 · 3 Mar 2020

Review

The manuscript describes the implementation of the trajectory-grid technique into the popular CMAQ regional air quality model. It demonstrates the use of this upgraded model in simulating transport of CO during the KORUS-AQ campaign, demonstrating that it could be used to identify likely sources of observed pollution.

The paper is fundamentally suitable for GMD, as it describes an interesting new diagnostic technique for a widely-used model. The trajectory-grid method is interesting, and the integration of a Lagrangian technique into an Eulerian model has the potential to bring valuable new insights into air quality modeling. However, I believe that the authors overstate their conclusions in ways which are not necessarily supported by their work. I also find that the paper lacks some important methodological details. I would therefore recommend that the paper undergo some revisions before being accepted for publication in GMD.

Major comments

My biggest concern is that certain aspects of the method seem to be either incompletely described or just incomplete. In particular:

1) If parcels are spawned in cells with the concentrations of the closest packet, how is this approach mass conservative? Chock et al (2005) did claim that the T-G method is mass conserving, but they point out that this is not the case when diffusion is simulated. Similarly, it is not clear to me how the spawning and pruning processes (lines 105-109) could occur without either incurring mass conservation errors or acting to artificially diffuse the concentration distribution.

2) The approach described is designed with long range transport in mind, but I could not find any description of how convection is treated. Convective transport could significantly change the path taken by an advected air parcel, and is not trivial to simulate in a Lagrangian framework. How is this phenomenon dealt with in this model? I could not see any reference to it in Figure 2.

I disagree with one of the premises of this study. On lines 46-48, it is claimed that back-trajectory modeling for source-receptor estimation is "not widely accepted because it is unable to determine whether an originated air mass is polluted or non-polluted". Given the widespread use of back-trajectory modeling for this exact purpose, this statement does not seem justified. Furthermore, it seems to neglect that the same issue exists with the C-TRAIL approach, but in reverse. Unless an simulated air mass exactly

coincides with the monitor location at the exact time of an observation, one cannot be certain that it contributed to that specific reading.

Line 198 – what does it mean to "reach Seoul" (altitude)? What is the minimum altitude that a packet has to reach to be considered as "reaching" Seoul? This seems important – if the altitude is too great, then air packets which are simply passing over Seoul (and therefore irrelevant to its air quality) will be incorrectly labeled as contributions. Furthermore it would be strange to compare two parcels which arrived at very different heights.

Throughout the paper, it is claimed that the model is "accurate" (line 296), "ideal" (line 302), and "validated" (line 11). However, the main validation is shown in Figure 4, and – unless I have misunderstood – this just shows that the base CMAQ model (since it does not seem that the TG modifications change the behavior of the base model) can reproduce some of the observed behavior. With that in mind:

1) Why does the model seem to consistently underpredict the observed values? I note that the authors do state as much for the DWP and SP, but they also state "very high correlation" during the EPP. While technically true that the index of correlation is higher during EPP than DWP or SP, this seems misleading when the root mean square error is still 68.7 ppbv.

2) Why does the model appear to have a hard minimum concentration of 50 ppbv CO (see figure 4a)?

3) In what sense is the TG approach "validated" here? If no specific validation of the C-TRAIL approach is given (e.g. by comparing to continuous satellite observations), I would recommend that the authors instead simply state that this is a new diagnostic technique for an existing model.

I applaud the authors for bringing up some of the shortcomings of the method in their conclusions (lines 305-306). However, I recommend that they put a more comprehensive discussion of these limitations in the main text.

Minor comments

Line 29 – the words in HYSPLIT should be capitalized (i.e. "Hybrid Single-Particle Lagrangian Integrated Trajectory")

The manuscript contains numerous grammatical errors. This does not affect the quality of the science or whether it should be accepted, but I would recommend that the authors take another pass through to try and improve this.

I noticed that the code is made available only by request. While this is fine, I would encourage the authors to consider making the code more freely available (e.g. on GitHub) unless they are unable to do so because of some specific restriction.

I recommend that the abstract be toned down. Several claims do not seem to be meaningful (e.g. on line 6, how is the Lagrangian output "reliable" or "comprehensive"?) and others are confusing (e.g. I'm not sure what is meant by "real polluted air masses" – line 8).

More generally, I note that the term "real polluted air mass" turns up many times. I suggest this is changed to simply "polluted air mass" given that these are still all simulation results. Even if the pollution was observed at some point, the model is performing a simulation which will include errors.

On lines 309 and 310, the authors call the model "efficient". However, I did not see any quantification of this in the paper. I suggest that this claim is removed. Alternatively the authors could consider giving a specific estimate of the computational overhead associated with C-TRAIL.

---

## Referee Comment (RC2) · Anonymous Referee #2 · 19 Mar 2020

Review This paper introduces and demonstrates how C-TRAIL can be used to enhance the value existing back trajectories calculation techniques for understanding long range transport of trace gases. In addition to the meteorological information that is already included to compute the trajectory, C-TRAIL incorporates trace gas concentration changes throughout the path of the trajectory. The authors applied this technique for the KORUS-AQ time period and demonstrated it's performance during various meteorological conditions. This new technique has the potential to be of high value for

scientist evaluating long range transport, scientists involved in model development, and air quality regulators. This paper fits the criteria for publication in GMD, but some methodological issues need to be addressed before accepting for publication.

Major comments

The authors make reference to "packets" starting on line 83 and the term is used throughout the paper. Is this considered to be a grid cell, a certain part of the concentration profile, the integrated profile or would the term air parcel be more appropriate? This should be clarified further because in some readers minds this term is unclear.

There is no mentioned about how well the meteorological model performs during the campaign period. Is there any previous work that compared surface and upper level winds to the model? During this campaign there were radiosondes launched (due to ozonesondes) that could provide some seamless validation of the model in the vertical and the usual stationary meteorological stations to verify the winds at least over land. I don't expect the authors to go ahead with a full scale meteorological validation, but I believe this is important to mention in more detail and include some references on the quality of wind outputs from WRF.

It is not clear what altitude these packets are arriving at over Seoul or if these packets started at various altitudes and descended to the surface via subsidence. This needs clarification considering this paper is discussing surface air quality impacts. This applies to lines 186-187, 21, and the discussion in lines 198-205. Additionally, while understanding the concentration change along the trajectory is a notable accomplishment of this study, the altitude of these packets needs to be included somehow in Figure 6b.

Lines 239-240: Wouldn't it be expected that almost all long distance packets would show lower concentrations because these are at higher altitudes, with stronger wind speeds, and not likely impacted from surface based CO sources? Are there convective processes considered in WRF that would transport surface CO at the altitudes considered here?

[Figure]

Line 254-255: Using the word "indisputable" is a strong statement and I believe needs some further justification. Since this application of C-TRAIL is based on understanding CO at the surface, it is important to know how well the modeled CO is relative to the surface measurements of CO. The model comparison to aircraft measurements gives an overview of the performance at numerous altitudes. However, since many of the high concentration CO packets remain close to the surface throughout their trajectory, it would be ideal to see how well the model performed when compared against surface monitor data.

Minor comments

There are some grammar issues that should be addressed in this paper. There were a few instances of present tense being used in combination with past tense and some sentences that should be restructured to better communicated the science presented in this paper. I suggest the authors give this a few reads to increase all the grammatical issues.

Line 63-66: In this sentence, it is not clear why the amount of pollution in an airmass would affect the reliability of the meteorology used in the trajectory calculation.

Line 138: A couple references to those papers would be ideal to include

Figure 4: the number of data points (n=##) should be included with these plots, especially to underscore the contrast between the number of samples for each of these conditions. This is especially relevant to the statement made on 148, where the highest correlation was found with the meteorological set up that had the least number of data points.

Line 158-159: It is not clear what point is being conveyed in this sentence. Isn't the sensitivity of CO source regions relevant for all conditions?

Line 162: I would suggest the complexity of the model is increased under stagnant conditions due to the lack of dispersion and one is further relying on getting the chemistry

right in the model. Please comment on this.

Figure 7-10: The plots in (b) and (c) should be expanded vertically and with x and y grids added to better see the changes in concentrations and distances. In some instances it is difficult to see the increases or decreases mentioned in the discussion.

Specifically in Figure 9, since most of the packets do not extend over a large distance, it would be helpful to have a more zoomed in map projection to show the detail in the path of these packets. This suggestion applies for the other figures (7-10). Additionally, if each of these packets that start at an altitude higher than the surface do indeed end up descending to the surface, another plot with the zoomed in domain over South Korea would be helpful for further understanding how these packets closer to the receptor site. Additionally, there are no units mentioned for the altitude colorbar in figures 7-10.

While the highlight of this paper is the role of LRT of pollution from China on Seoul air quality, it would be helpful to have a zoomed in view of just South Korea to see the variability of surface packets.

Line 248: The text says Figure 9, but this should be Figure 10.

Figure 12a and 12b: Why are some of the clusters partially colored or not colored at all? Visually, it would be ideal to have all of them colored so it is obvious which cluster is which.

Line 264-266:

---

## Author Comment (AC1) · 18 Apr 2020

**Response to reviewer #1**

Authors appreciate reviewer's thoughtful comments and suggestions, which are greatly helpful for us to improve our manuscript. The manuscript has been revised to accommodate the reviewer's comments.

**General Comments:**

**The manuscript describes the implementation of the trajectory-grid technique into the popular CMAQ regional air quality model. It demonstrates the use of this upgraded model in simulating transport of CO during the KORUS-AQ campaign, demonstrating that it could be used to identify likely sources of observed pollution. The paper is fundamentally suitable for GMD, as it describes an interesting new diagnostic technique for a widely-used model. The trajectory-grid method is interesting, and the integration of a Lagrangian technique into an Eulerian model has the potential to bring valuable new insights into air quality modeling. However, I believe that the authors overstate their conclusions in ways which are not necessarily supported by their work. I also find that the paper lacks some important methodological details. I would therefore recommend that the paper undergo some revisions before being accepted for publication in GMD.**

**Major comments: My biggest concern is that certain aspects of the method seem to be either incompletely described or just incomplete. In particular:**

**Comment) If parcels are spawned in cells with the concentrations of the closest packet, how is this approach mass conservative? Chock et al (2005) did claim that the T-G method is mass conserving, but they point out that this is not the case when diffusion is simulated. Similarly, it is not clear to me how the spawning and pruning processes (lines 105-109) could occur without either incurring mass conservation errors or acting to artificially diffuse the concentration distribution.**

**Response)** Thank you for your insightful comment. As we mentioned in the manuscript, Line 96-99: "Since we can use the TG method to calculate the concentration from an ordinary differential equation, it is mass conserving, monotonic, and accurate. In the diffusion step, however, interpolation errors occur, but they are typically considerably smaller than Eulerian advection errors (Chock et al., 2005)." Therefore, your comment is valid, and we addressed the error caused by pruning and spawning by adding another sentence in Line 125-126 "The limitation of this packet management approach, however, is that it incurs mass conservation by adding minor interpolation errors."

**Comment) The approach described is designed with long range transport in mind, but I could not find any description of how convection is treated. Convective transport could significantly change the path taken by an advected air parcel, and is not trivial to simulate in a Lagrangian framework. How is this phenomenon dealt with in this model? I could not see any reference to it in Figure 2.**

**Response)** Based on CMAQ's Science document, Chapter 11: "The resolved clouds have been simulated by the MM5 to cover the entire grid cell. No additional cloud dynamics are considered in CMAQ for this cloud type since any convection and/or mixing would have been resolved and considered in the vertical wind fields provided by MM5." Also, we checked the WRF Kain-Fritsch Model: "…, momentum transport in convective clouds is crudely simulated by assuming conservation of momentum in convective clouds…"

$$\frac{\Delta \bar{u}}{\Delta t}\bigg|_{conv} = \frac{1}{\Delta p}[(\omega_{u2} + \omega_{d2})\bar{u}_2 - (\omega_{u1} + \omega_{d1})\bar{u}_1 + (\epsilon_u + \epsilon_d)\bar{u}_m - \delta_u u_{um} - \delta_d u_{dm}]$$

$$\frac{\Delta \bar{v}}{\Delta t}\bigg|_{conv} = \frac{1}{\Delta p}[(\omega_{u2} + \omega_{d2})\bar{v}_2 - (\omega_{u1} + \omega_{d1})\bar{v}_1 + (\epsilon_u + \epsilon_d)\bar{v}_m - \delta_u v_{um} - \delta_d v_{dm}]$$

TG advection in C-TRAIL solves the transport equation in a three-dimensional format; therefore, the convective transport is already considered in vertical wind fields. Sub-grid clouds convection, however, will be considered in a future study.

**Comment) I disagree with one of the premises of this study. On lines 46-48, it is claimed that backtrajectory modeling for source-receptor estimation is "not widely accepted because it is unable to determine whether an originated air mass is polluted or non-polluted". Given the widespread use of back-trajectory modeling for this exact purpose, this statement does not seem justified. Furthermore, it seems to neglect that the same issue exists with the C-TRAIL approach, but in reverse. Unless an simulated air mass exactly coincides with the monitor location at the exact time of an observation, one cannot be certain that it contributed to that specific reading.**

**Response)** Thank you for your comment. You are right about the widespread use of back-trajectory modeling for finding the originated air mass. In this study, however, our approach is not based on observations. For the domain of our study, we are using an emission inventory platform for a gridded emission inventory. With the help of this gridded domain and use of the TG method in following air parcels, we are attempting to link the receptor air parcel with the originated grid. Therefore, we determine how much pollution the originated grid provided by its emission and concentrations. The C-TRAIL approach is a forward trajectory model, with millions of air parcels moving in the domain, some of which will pass through (or land in) the receptor location. Unlike other conventional trajectory methods, this approach uses gridded emissions/concentrations across the domain (which is not the case for observation sites; that is, they are available only where the observation site is located). Therefore, we think that the C-TRAIL approach has some benefits. The limitation of this approach, however, is the uncertainty of the emission inventory and/or model concentrations (which is why we provided the correlation and IOA values).

**Comment) Line 198 – what does it mean to "reach Seoul" (altitude)? What is the minimum altitude that a packet must reach to be considered as "reaching" Seoul? This seems important – if the altitude is too great, then air packets which are simply passing over Seoul (and therefore irrelevant to its air quality) will be incorrectly labeled as contributions. Furthermore it would be strange to compare two parcels which arrived at very different heights.**

**Response)** Thank you for your comment. The word "reaching," as you mentioned, does not always mean affecting air quality issues, but as we have shown in the supplemental Figure S8 referring to heights, the air parcels over Seoul are at the 1.5km and lower more than 80% of the time. Furthermore, in our explanations, we divided the air parcels that come from higher altitudes from those which are at a lower altitude (below the boundary layer height ~ 1km) to determine the impact on LRT. You can also find this information in Figures 7-10 (the color-bar), in which most of the air parcels are blue. We agree with your point, so we made some modifications to the manuscript.

Line 199-200: "… different packets from different altitudes (from below 1 km to almost 10 km)… "

Line 226: "**Error! Reference source not found.**(a) presents the general path of all packet trajectories reaching the Seoul area at different altitudes at 9:00 AM LT throughout May 2016."

Line 215-216: "The concentrations of near-surface packets tend to fluctuate more than those of high-altitude packets (Figure S7)."

We also added a plot in a supplementary document for height change of 4 packets reaching Seoul to show their altitudes:

[Figure]

Figure S7: Changes in the height (km) of four aged packets moving toward Seoul from source points

**Comment) Throughout the paper, it is claimed that the model is "accurate" (line 296), "ideal" (line 302), and "validated" (line 11). However, the main validation is shown in Figure 4, and – unless I have misunderstood – this just shows that the base CMAQ model (since it does not seem that the TG modifications change the behavior of the base model) can reproduce some of the observed behavior. With that in mind: 1) Why does the model seem to consistently underpredict the observed values? I note that the authors do state as much for the DWP and SP, but they also state "very high correlation" during the EPP. While technically true that the index of correlation is higher during EPP than DWP or SP, this seems misleading when the root mean square error is still 68.7 ppbv. 2) Why does the model appear to have a hard minimum concentration of 50 ppbv CO (see figure 4a)? 3) In what sense is the TG approach "validated" here? If no specific validation of the C-TRAIL approach is given (e.g. by comparing to continuous satellite observations), I would recommend that the authors instead simply state that this is a new diagnostic technique for an existing model. I applaud the authors for bringing up some of the shortcomings of the method in their conclusions (lines 305-306). However, I recommend that they put a more comprehensive discussion of these limitations in the main text.**

Response) 1) The underprediction of the standard CMAQ model vs. observations is mostly due to the quality of the emission inventory over East Asia. Several studies have obtained similar results from CMAQ over East Asia. When providing the correlation, however, we also calculate the IOA (index of agreement), which combines the correlation and mean bias errors. The IOA for EPP shows a 0.88 value, which is generally high for the IOA. Your concern regarding the underprediction will be addressed by data assimilation and inverse modeling techniques in future studies.

2) The model here is compared to the KORUS-AQ aircraft observation datasets provided in the KORUS-AQ website. The dataset covers several species at different altitudes during the KORUS-AQ period, May and June 2016. The minimum of about 50ppbv comes from observations that established this minimum from measurements. We also compared our model's CO predictions with surface measurements for stations across South Korea (Table S3, Figure S5). The results showed the discussed underprediction, which we think is because of uncertain data acquisition for ground measurements (error ±100 ppbv for CO), uncertain emission inventory over East Asia, and model's uncertainty in chemistry modeling.

3) We agree with the reviewer that this model is a new diagnostic technique for the CMAQ model, and we added a related comment in the Discussion section. In the following version of C-TRAIL in a future study, we will implement all CMAQ modules in C-TRAIL and compare these two models with observations.

Line 11-12: "The diagnostic output from C-TRAIL accurately identifies the origins of pollutants."

Line 190-191: "Because C-TRAIL is a diagnostic tool derived from CMAQ, both a Lagrangian output and CMAQ standard Eulerian output are available after each run."

**Comment) Minor comments Line 29 – the words in HYSPLIT should be capitalized (i.e. "Hybrid Single-Particle Lagrangian Integrated Trajectory")**

**Response)** Thanks for your comment. Revised.

**Comment) The manuscript contains numerous grammatical errors. This does not affect the quality of the science or whether it should be accepted, but I would recommend that the authors take another pass through to try and improve this.**

**Response)** We rewrote some sections due to some grammatical errors. Thank you for your suggestion.

**Comment) I noticed that the code is made available only by request. While this is fine, I would encourage the authors to consider making the code more freely available (e.g. on GitHub) unless they are unable to do so because of some specific restriction.**

**Response)** Based on reviewer's suggestion we provided this code in Github. You can find the link at the Code Availability section.

Line 348-349: "**Code Availability.** The C-TRAIL version 1.0 is available for non-commercial research purposes at https://github.com/armanpouyaei/C-TRAIL-v1.0."

**Comment) I recommend that the abstract be toned down.**

**Response)** Thanks, Revised.

**Comment) Several claims do not seem to be meaningful (e.g. on line 6, how is the Lagrangian output "reliable" or "comprehensive"?) and others are confusing (e.g. I'm not sure what is meant by "real polluted air masses" – line 8).**

**Response)** We addressed these types of claims in the manuscript. Thanks.

**Comment) More generally, I note that the term "real polluted air mass" turns up many times. I suggest this is changed to simply "polluted air mass" given that these are still all simulation results. Even if the pollution was observed at some point, the model is performing a simulation which will include errors.**

**Response)** Addressed.

Line 8: "…for validating the source-receptor direct link by following polluted air masses."

Line 63: ": (1) to provide a direct link between polluted air masses from sources and a receptor"

**Comment) On lines 309 and 310, the authors call the model "efficient". However, I did not see any quantification of this in the paper. I suggest that this claim is removed. Alternatively the authors could consider giving a specific estimate of the computational overhead associated with C-TRAIL.**

**R)** Thanks for the comment. Based on your suggestion, we removed this claim.

---

## Author Comment (AC2) · 18 Apr 2020

**Response to reviewer #2**

Authors appreciate reviewer's thoughtful comments and suggestions, which are greatly helpful for us to improve our manuscript. The manuscript has been revised to accommodate the reviewer's comments.

**General Comment:**

**This paper introduces and demonstrates how C-TRAIL can be used to enhance the value existing back trajectories calculation techniques for understanding long range transport of trace gases. In addition to the meteorological information that is already included to compute the trajectory, C-TRAIL incorporates trace gas concentration changes throughout the path of the trajectory. The authors applied this technique for the KORUS-AQ time period and demonstrated it's performance during various meteorological conditions. This new technique has the potential to be of high value for scientist evaluating long range transport, scientists involved in model development, and air quality regulators. This paper fits the criteria for publication in GMD, but some methodological issues need to be addressed before accepting for publication.**

**Major comments**

**COMMENT) The authors make reference to "packets" starting on line 83 and the term is used throughout the paper. Is this considered to be a grid cell, a certain part of the concentration profile, the integrated profile or would the term air parcel be more appropriate? This should be clarified further because in some readers minds this term is unclear.**

**RESPONSE)** Thanks for your comments. In Lagrangian format, we needed to work with particles rather than grid cells, and these particles were representative of air parcels. As their nature, however, differs from that of air parcels and they carry several species, we referred to them as "air packets." Based on your concern, we modified the text to elaborate on the definition of "packets" as follows:

Line 80-82: "From now on, we will call these "points" air packets, because (1) their nature is similar to air parcels, but they are much smaller in size, (2) they behave like particles, but they are carrying information for several species."

**COMMENT) There is no mentioned about how well the meteorological model performs during the campaign period. Is there any previous work that compared surface and upper level winds to the model? During this campaign there were radiosondes launched (due to ozonesondes) that could provide some seamless validation of the model in the vertical and the usual stationary meteorological stations to verify the winds at least over land. I don't expect the authors to go ahead with a full scale meteorological validation, but I believe this is important to mention in more detail and include some references on the quality of wind outputs from WRF.**

**RESPONSE)** Thanks for your suggestion. Following your suggestion, we did some validation for horizontal wind over surface. In addition, we downloaded Sondes data from the KORUS-AQ website and prepared some validation on the vertical profile of wind. We presented the results here and in the revised manuscript and added the plots to the supplement of the manuscript (Table S1-S2, Figure S1-S4).

Line 140-142: "We validate our WRF model's wind predictions with surface measurements and radiosonde measurements for the KORUS-AQ period (see Table S1-S2 and Figure S1-S4)."

U and V validation based on surface stations (255 stations across South Korea):

| | IOA | Correlation | RMSE | MAE |
|---|---|---|---|---|
| **U wind velocity** | 0.7632 | 0.6927 | 2.1952 | 1.7536 |
| **V wind velocity** | 0.6922 | 0.5663 | 2.1924 | 1.7282 |

[Figure]

[Figure]

Radiosonde validation for wind speed and direction (82 observation point for hourly averaged sonde data):

| | IOA | Correlation | RMSE | MAE |
|---|---|---|---|---|
| **Wind Speed** | 0.8616 | 0.7642 | 6.5722 | 4.9589 |
| **Wind Direction** | 0.7341 | 0.6269 | 70.1072 | 49.2412 |

[Figure]

**COMMENT) It is not clear what altitude these packets are arriving at over Seoul or if these packets started at various altitudes and descended to the surface via subsidence. This needs clarification considering this paper is discussing surface air quality impacts. This applies to lines 186-187, 21, and the discussion in lines 198-205.**

RESPONSE) Thank you for your comment. The information regarding the altitude of these packets is provided in the color-bar of Figures 7-10 and extra box-plots in the supplementary document (Figure S8). We clarified your points in line 199-202 of the revised manuscript: "This section provides an example of how we use C-TRAIL to study the sources of different packets from different altitudes (from below 1 km to almost 10 km) over the Seoul Metropolitan Area (SMA); later sections will focus on the entire month of May 2016 C-TRAIL over the SMA and provide more comprehensive illustrations of the concentrations and altitudes of trajectories."

**COMMENT) Additionally, while understanding the concentration change along the trajectory is a notable accomplishment of this study, the altitude of these packets needs to be included somehow in Figure 6b.**

RESPONSE) Thank you for your comment. We added a figure showing how the altitude of these four packets change on this day in the supplement (Figure S7).

[Figure]

Figure S7: Changes in the height (km) of four aged packets moving toward Seoul from source points

Also, we have added altitude graphs in the supplement for Figures 7-10 (Figure S8).

**COMMENT) Lines 239-240: Wouldn't it be expected that almost all long distance packets would show lower concentrations because these are at higher altitudes, with stronger wind speeds, and not likely impacted from surface based CO sources? Are there convective processes considered in WRF that would transport surface CO at the altitudes considered here?**

**RESPONSE)** Thank you for raising this concern. Your assumption is valid, but some convective transport results from WRF convection, which could directly impact vertical wind and therefore the surface CO concentration. The impact, however, is not as great as that of horizontal transport. We addressed this concern in response to a comment by reviewer #1 below: Based on the CMAQ Science document, Chapter 11: "The resolved clouds have been simulated by the MM5 to cover the entire grid cell. No additional cloud dynamics are considered in CMAQ for this cloud type since any convection and/or mixing would have been resolved and considered in the vertical wind fields provided by MM5." Also, we checked WRF Kain-Fritsch Model: "…, momentum transport in convective clouds is crudely simulated by assuming conservation of momentum in convective clouds…"

$$\frac{\Delta \bar{u}}{\Delta t}\bigg|_{conv} = \frac{1}{\Delta p}[(\omega_{u2} + \omega_{d2})\bar{u}_2 - (\omega_{u1} + \omega_{d1})\bar{u}_1 + (\epsilon_u + \epsilon_d)\bar{u}_m - \delta_u u_{um} - \delta_d u_{dm}]$$

$$\frac{\Delta \bar{v}}{\Delta t}\bigg|_{conv} = \frac{1}{\Delta p}[(\omega_{u2} + \omega_{d2})\bar{v}_2 - (\omega_{u1} + \omega_{d1})\bar{v}_1 + (\epsilon_u + \epsilon_d)\bar{v}_m - \delta_u v_{um} - \delta_d v_{dm}]$$

TG advection in C-TRAIL solves the transport equation in three-dimensional format, therefore, the convective transport is already considered in vertical wind fields. Sub-grid cloud convection, however, will be considered in a future study.

**COMMENT) Line 254-255: Using the word "indisputable" is a strong statement and I believe needs some further justification.**

**RESPONSE)** Thank you for your comment. To clarify for readers, we changed the wordings of this sentence:

Line 272-273: "That is, the findings of this study regarding the trajectories and the origin of polluted air masses are similar to those of previous studies."

**COMMENT) Since this application of C-TRAIL is based on understanding CO at the surface, it is important to know how well the modeled CO is relative to the surface measurements of CO. The**

**model comparison to aircraft measurements gives an overview of the performance at numerous altitudes. However, since many of the high concentration CO packets remain close to the surface throughout their trajectory, it would be ideal to see how well the model performed when compared against surface monitor data.**

**RESPONSE)** Thank you for your comment. Since the data available for CO measurement over the surface has high uncertainties (±100 ppbv), we did a brief validation for 25 stations over South Korea. Another major issue is the underprediction of the model due to the high uncertainty in emission inventory over East Asia that has been discussed in several previous papers. Since this paper is proposing a new diagnostic tool based on CMAQ, as you mentioned, it is more valuable if it could be applied over a region with a CMAQ model with high performance. Anyhow, below you can find CO surface measurements validation. We also added this result into supplementary document (Table S3, Figure S5)

| | IOA | Correlation | RMSE | MAE |
|---|---|---|---|---|
| **CO concentration** | 0.4932 | 0.3128 | 199.85 | 168.02 |

[Figure]

In the future study, we will apply some data assimilation techniques and artificial intelligence bias correction methods to improve the CMAQ results over this region. We also revised the manuscript for mentioning this issue:

Line 157-160: "We also provided CMAQ's CO comparison with surface stations measurements in the supplementary document (see Table S3 and Figure S5). The results of this comparison also show the model's underprediction, which is caused by uncertain emission inventories over East Asia."

**Minor comments**

**COMMENT) There are some grammar issues that should be addressed in this paper. There were a few instances of present tense being used in combination with past tense and some sentences that should be restructured to better communicated the science presented in this paper. I suggest the authors give this a few reads to increase all the grammatical issues.**

**RESPONSE)** Thank you for sharing your concern. We have proofread the manuscript several times and changed some wording because of grammatical issues in the text. You will see these changes in the revised draft of the manuscript.

**COMMENT) Line 63-66: In this sentence, it is not clear why the amount of pollution in an airmass would affect the reliability of the meteorology used in the trajectory calculation.**

**RESPONSE)** Thank you for your comment. As you stated, the sentence was vague, so we rephrased it as follows:

Line 66-68: "One significant outcome of the TG model applied to CTMs is its ability to account for the concentrations of pollutants in air masses in its investigation of trajectories, which addresses the unreliability of meteorology-based Lagrangian models when the pollutedness or cleanliness of an originated air mass becomes an issue."

**COMMENT) Line 138: A couple references to those papers would be ideal to include**

**RESPONSE)** Thanks for your comment. The following references are some that we added to the revised manuscript:

Line 146-147: "In papers pertaining to this campaign, several studies (Al-Saadi et al., 2016; Choi et al., 2019; Miyazaki et al., 2019) separated the time frame into three periods (Table 1)"

**COMMENT) Figure 4: the number of data points (n=##) should be included with these plots, especially to underscore the contrast between the number of samples for each of these conditions. This is especially relevant to the statement made on 148, where the highest correlation was found with the meteorological set up that had the least number of data points.**

**RESPONSE)** Thank you for pointing out this thoughtful point. As you mentioned, we added the number of data points in the caption of the plot.

Entire May 2016: n=6865

DWP: n=1750

SP: n=1548

EPP: n=264

**COMMENT) Line 158-159: It is not clear what point is being conveyed in this sentence. Isn't the sensitivity of CO source regions relevant for all conditions?**

**RESPONSE)** Thank you for your comment. Here, we have stated that the vertical velocities associated with convection could have significantly impacted transport for DWP period (which consisted of rainy and cloudy days). To convey this point, we rephrased our sentence as follows:

Line 169-170: "Also, in light of the impact of convection, the concentrations of CO over Korea could have increased or decreased by vertical wind transport and cloud updrafts and downdrafts."

**COMMENT) Line 162: I would suggest the complexity of the model is increased under stagnant conditions due to the lack of dispersion and one is further relying on getting the chemistry right in the model. Please comment on this.**

**RESPONSE)** Thank you for this thoughtful suggestion. We agree with the reviewer and revised our sentences as follows:

Line 173-176: "Even though one might assume that the model would produce more accurate simulations with less convection-related transport, CO concentrations were significantly underestimated by the model

(Jeon et al., 2016) because of uncertainties in the chemistry modeling and the faulty emission inventories over East Asia."

**COMMENT) Figure 7-10: The plots in (b) and (c) should be expanded vertically and with x and y grids added to better see the changes in concentrations and distances. In some instances it is difficult to see the increases or decreases mentioned in the discussion. Specifically in Figure 9, since most of the packets do not extend over a large distance, it would be helpful to have a more zoomed in map projection to show the detail in the path of these packets. This suggestion applies for the other figures (7-10). Additionally, if each of these packets that start at an altitude higher than the surface do indeed end up descending to the surface, another plot with the zoomed in domain over South Korea would be helpful for further understanding how these packets closer to the receptor site. Additionally, there are no units mentioned for the altitude colorbar in figures 7-10. While the highlight of this paper is the role of LRT of pollution from China on Seoul air quality, it would be helpful to have a zoomed in view of just South Korea to see the variability of surface packets.**

**RESPONSE)** Thanks. We addressed all of your concerns regarding Figures 7-10. We revised the plots and added a zoomed map over South Korea for each period. You can find the revised Figure 7 below:

[Figure]

(d)

[Figure]

Figure 1: C-TRAIL output for the entire month of May 2016 for Seoul as the receptor: (a) 24-hour trajectories of packets for the entire domain, (b) 24-hour trajectories of packets for the zoomed area in South Korea, (c) boxplot of the CO concentrations of all packets at each hour before they reached Seoul, and (d) the boxplot of packet distances from Seoul every hour before the packets reached Seoul

**COMMENT) Line 248: The text says Figure 9, but this should be Figure 10.**

**RESPONSE)** Thank you, Revised.

**COMMENT) Figure 12a and 12b: Why are some of the clusters partially colored or not colored at all? Visually, it would be ideal to have all of them colored so it is obvious which cluster is which.**

**RESPONSE)** Thanks. Revised.

---

## Author Comment (AC3) · 18 Apr 2020

The comment was uploaded in the form of a supplement: https://www.geosci-model-dev-discuss.net/gmd-2019-366/gmd-2019-366-AC3-supplement.pdf

---

## Author Response (AR2)

**Response to minor revisions from Anonymous Referee #1**

Authors appreciate reviewer's thoughtful comments and suggestions. The manuscript has been revised to accommodate the reviewer's comments.

**General Comments:**

**I thank the authors for their responses to my comments. However, some issues remain. I have limited myself to issues which came up during my own prior review. I believe that this manuscript can in principle be published, but that some claims made in the paper still need to be moderated, and some additional details provided.**

**Comment) While I agree that C-TRAIL seems like a useful tool and look forward to seeing it applied more broadly, there does not seem to be sufficient quantitative evidence for claims in the abstract and elsewhere that the model can be called "reliable" and "comprehensive". A true assessment of reliability would require comparison of C-TRAIL to results from other models for many different scenarios, rather than a single campaign. Furthermore, C-TRAIL has its own issues, including mass conservation errors and the (ubiquitous) uncertainty in meteorology and emissions inventories. I would recommend that the authors remove words like "reliable", and "comprehensive" from the abstract, as these claims are not meaningfully quantified in the paper.**

**Response)** Thanks for your comments. We removed "reliable" and "comprehensive" as the reviewer suggested. The reason why we called our model "reliable" was that for this case study (KORUS-AQ), it showed somehow similar conclusions to previous studies. However, in case of being reliable as it can be applied for many different campagins and/or scenarios, your statement is right. Also, the claim of "comprehensive" was due to model's capability in seeing different aspects in a source-receptor problem, which as you mentioned may not be the perfect adjective to use here. Again, we removed "reliable" and "comprehensive" and added "novel" to represent the novelty of the output.

**Comment) On a similar note, it seems too strong to claim that any model is "ideal" (line 342).**

**Response)** Thanks. We replaced "ideal" with "practical".

**Comments) The conclusions still state that C-TRAIL is highly computationally efficient (line 348), but this is not quantified in the manuscript. Either this claim should be removed or a quantitative assessment of computational overhead due to C-TRAIL should be included.**

**Response)** As the reviewer suggested, we removed our claim.

**Comment) The authors have added a description of air "packets", which helps considerably. However, it would be useful to clarify that the packets are massless (if they are, as I assume). If they are not massless, then the question remains of how they can be pruned and spawned without incurring additional errors in mass conservation, beyond the "interpolation" errors already discussed by the authors. If they are indeed massless, then they also presumably have no size, rather than being simply small (lines 80-82).**

**Response)** Thanks for the comment. As the reviewer addressed, packets don't have any mass. So, we modified the sentence into:

"we will refer to these points as "air packets" for two reasons: (1) Their nature is similar to that of air parcels, but they are massless, and (2) they behave much like particles, but they carry information about several species."

**Comment) I appreciated the authors' response regarding issues modeling cloud convection. However, this hinges on the approach taken by WRF and CMAQ for simulating convection, as well as the resolution of the original WRF simulation. It would be helpful to the reader if the authors included a technical description of how each of the three components (WRF, CMAQ, and C-TRAIL) attempts to account for convection. This would help both in understanding limitations of the approach and in ensuring that readers can reproduce the results.**

**Response)** Thanks for the suggestion. We added a table in a supplementary document and the below paragraph into the revised manuscript:

"In addition, the C-TRAIL model considers convective transport only for resolved clouds (when clouds cover an entire grid). The WRF model implements the cloud model to obtain cloud properties on a sub-grid scale and addresses vertical transport on a resolved scale (Kain, 2004). Outputs from the WRF model are used in the CMAQ's cloud model to account for convective transport in two separate modules: sub-grid scale clouds and resolved clouds (Byun and Schere, 2006). In this version of C-TRAIL for convection, we only use vertical winds determined from resolved clouds (see Table S1)."

**Table S1: Convective transport treatment in WRF, CMAQ, and C-TRAIL**

| Models | WRF v3.8 | | CMAQ v5.2 | | C-TRAIL v1.0 | |
|---|---|---|---|---|---|---|
| **Convection Scheme** | Kain-Fritch (2004) | | ACM | | ACM modified | |
| **Scales** | Resolved | Sub-grid | Resolved | Sub-grid | Resolved | Sub-grid |
| **Availabilty** | ✓ | ✓ | ✓ | ✓ | ✓ | ✕ |

**Comment) There remain some grammatical errors, which unfortunately harm the readability of the paper. For example, line 125 reads "it incurs mass conservation by adding minor interpolation errors", but this doesn't make sense (I assume this was meant to be "it violates mass conservation" or "it incurs mass conservation errors"?). I recommend that authors perform an additional sweep for such issues, as well as fixing some of the figure references which appeared as "Error! Reference source not found".**

**Response)** Thanks for the suggestion. In addition to a professional editor, we proofread the paper and revised for grammatical and reference-related errors.

---

## Author Response (AR3)

**Response to final comments from Editor**

Authors appreciate editor's comments and suggestions regarding code and data availability. The manuscript has been revised to accommodate the editor's comments.

**General Comments:**

**Thanks you for the revision. I am happy to accept them as is. We had a further look at the GMD data policy. Here are some comments, which I like you to have a close look at and revise the data handling accordingly.**

**COMMENT 1) Github URLs. Github is an excellent development platform, but it lacks the features required of an archive. GitHub themselves tell authors to use Zenodo for this purpose. The authors should follow the procedure detailed there to archive the exact version of the software used to create the results presented: https://guides.github.com/activities/citable-code/. The resulting Zenodo repositories present the correct bibliography entries to use.**

**RESPONSE)** Thanks, based on your request, a Zenodo repository was created for C-TRAIL v1.0 with specific DOI. This was mentioned in the section of Code and Data availablitity:

"The C-TRAIL version 1.0 is derived from the CMAQ v5.2 model. The CMAQ model is an open-source model that can be accessed via Zenodo (https://doi.org/10.5281/zenodo.1167892). The documentation and tutorials on CMAQ are available on the US EPA modeling website https://www.cmascenter.org/ (last access: June 2020). The C-TRAIL v1.0 model source code and pre/post-processing scripts are available via Zenodo (https://doi.org/10.5281/zenodo.3885782)."

**COMMENT 2) No data identified. The datasets used to conduct the evaluation experiments presented must be cited from the code and data availability section with enough precision to allow a reader to reproduce the work in the manuscript.**

**RESPONSE)** We mentioned about data that was used for evaluations in the section of code and data availability:

"The flight observations used for evaluation of the model were downloaded from the KORUS-AQ website https://www-air.larc.nasa.gov/missions/korus-aq/ (last access: June 2020), DC8 Aircraft observations for CO concentrations comparison and Ozonsonde data for wind speed/direction evaluation. For surface concentration evaluations, we used surface observational data from the Air Quality Monitoring Station (AQMS) network operated by the National Institute of Environmental Research (NIER), which is available for the public on https://www.airkorea.or.kr/web (last access: June 2020)."

**COMMENT 3) No configuration, run, or data processing scripts. The configuration files, run scripts and any data processing or analysis scripts used to produce the results presented in the manuscript need to be publicly and persistently archived, and cited from the code and data availability section. As a guide, every file the user would need to reproduce the manuscript should accessible.**

**RESPONSE)** Thanks for your comment. The data processing and plotting scripts are also added to the C-TRAIL's Zenodo repository. For clustering and trajectory plots, we used an Open-air package in the R programming language. The library set up and run scripts for C-TRAIL are completely similar to the CMAQ model, and the process to configure the model and run is mentioned in README of the repository. I have to add that the configurations for the WRF run and Emission Inventory are also mentioned in the

manuscript, which can be used to reproduce the data as input for the CMAQ model. Once all the required input data for CMAQ is prepared, running C-TRAIL would be the same as running the CMAQ model.